# REVISITING THE SCALING EFFECTS OF LLMS ON MEDICAL REASONING CAPABILITIES

## ABSTRACT

Recently, LLMs such as the Llama and Qwen families have rapidly improved by significantly scaling their training corpora, with smaller models trained on larger datasets now approaching or surpassing the performance of previous-generation larger models on public benchmarks. In this paper, we revisit the scaling effects of LLMs, using the medical field as a case study, by carefully analyzing how training corpus size and parameter size affect model performance on problems of varying difficulty. To this end, we present MedResEval, a new benchmark built upon the MedQA dataset. It is designed to demand more complex reasoning and decision-making and more accurately reflect real-world medical scenarios. Leveraging MedResEval, we investigate the scaling effects of training corpus and model size in LLMs through a comprehensive analysis of several prominent LLM families on medical reasoning tasks of varying complexity. The results reveal that while smaller models like Llama 3 (8B) approach the performance of older, larger models like Llama 2 (70B) on simple tasks like MedQA, they consistently underperform on complex tasks requiring advanced reasoning. Furthermore, we develop a difficulty-dependent scaling-law formula to characterize how LLMs' performance varies with training data size at a fixed model parameter size. The formula reveals that reasoning error reduction rates are 1.3 times greater for large-scale LLMs ($\approx$ 70B) compared to small-scale LLMs ($\leq$10B) on simple tasks, and 2 times greater on difficult reasoning tasks. Our study highlights that while both data and parameter scales enhance LLM performance, greater emphasis must be placed on parameter scales, particularly for complex reasoning tasks. Only LLMs with sufficiently large parameters can effectively tackle the complexities of real-world medical scenarios.

## 1 INTRODUCTION

Large language model (LLM) technology has witnessed rapid advancement (Ouyang et al., 2022; Achiam et al., 2023; Anil et al., 2023; Touvron et al., 2023a) and shown potential in various domains. Scaling laws (Kaplan et al., 2020) highlight the critical role of training data size and model parameters in enhancing large language models (LLMs). Guided by scaling laws, mainstream LLMs families have been advancing new iterations by continuously scaling their training corpora. For example, the Llama family has evolved from the first generation, Llama 1, to the third generation, Llama 3, by expanding the training corpus size by about 10 times. Another well-known open-source LLM family Qwen has done something similar. Recently, LLMs have achieved significant advancements by significantly scaling their training corpora. The latest small-scale LLMs (Dubey et al., 2024; Hui et al., 2024) trained on larger datasets are now approaching or even surpassing the performance of previous large-size LLMs across various public benchmarks. For example, on the famous MMLU benchmark (Hendrycks et al.), the latest Llama3 (8B) (Dubey et al., 2024) trained with 15T tokens has significantly outperformed Llama1 (65B) (Touvron et al., 2023a) trained with 1.4T tokens. Therefore, a critical question arises that requires serious consideration: is it sufficient to solely scale up the training corpus to develop strong small-scale LLMs with capabilities of addressing various general or field-specific real-world application needs?

In this study, we revisit the scaling effects of both training corpus size and model parameters, examining how they influence the capabilities of LLMs. Since the medical field is a critical application area for LLMs, we use it as a case study to investigate the scaling effects. To ensure consistency

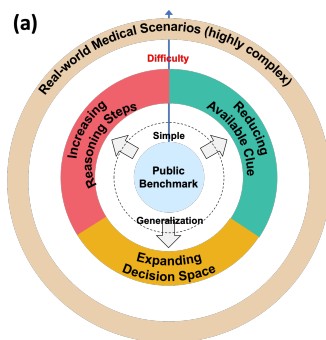 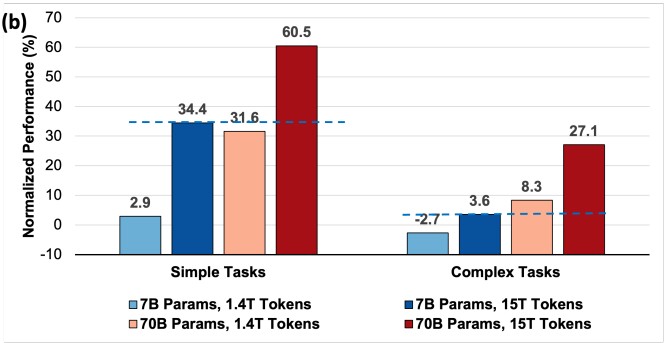

Figure 1: (a) The process of generating medical tasks corresponding to different generalization types based on existing simple evaluation tasks. (b) Performance of models (Llama) with varying numbers of parameters and training tokens on different levels of medical tasks. A performance score below 0 indicates that the model is performing worse than random guessing.

and comparability with previous work in the field, we also select the well-known public benchmark MedQA (Jin et al., 2021) as our initial evaluation dataset. However, in real clinical scenarios, medical reasoning problems are not about choosing the best answer from five predefined options, as in MedQA, but involve more complex situations—such as making correct decisions without predefined choices or requiring additional reasoning steps to reach an effective conclusion. For example, before making a disease diagnosis, clinicians must first identify and correct potentially inaccurate clinical information, which may stem from the patient's incomplete or inaccurate narratives or from erroneous records. Therefore, we further build a new benchmark MedResEval based on MedQA to investigate the scaling effects of LLMs. It is designed to demand more complex medical reasoning or decision-making and more reflect real-world medical scenarios by the following three typical manners (see Figure 1(a)) : (1) increasing reasoning difficulty by reducing available clues (such as without predetermined options), (2) expanding the decision space, and (3) increasing the reasoning steps needed to reach an effective decision. We examine the scaling effects of training corpus and model size in LLMs through a comprehensive analysis of several prominent LLM families by leveraging the proposed new evaluation benchmark MedResEval.

The experimental results demonstrate that while some of the latest small-scale LLMs like Llama 3 (8B) have achieved comparable or even superior performance to the older version large-scale LLMs on simple tasks like MedQA, they consistently underperform on difficult reasoning tasks that demand more advanced decision-making capabilities (see Figure 1(b)). Furthermore, we develop a difficulty-dependent scaling-law formula which can characterize LLMs' performance with varying training corpus at a fixed model parameter size. The formula quantitatively reveals that reasoning error reduction rates are 1.3 times greater for large-scale LLMs ($\approx$ 70B) compared to small-scale LLMs ($\leq$10B) on simple reasoning tasks, and 2 times greater on complex reasoning tasks. Our study highlights that while both data and parameter scales enhance LLM performance, greater emphasis must be placed on model parameter scales, particularly for difficult reasoning tasks, such as some real-world medical problems encountered. LLMs with sufficiently large parameters are essential for effectively addressing difficult reasoning problems. Finally, we further validated our findings regarding the parameter scale of LLMs by performing a diagnosis simulation from a real medical scenario.

## 2 RELATED WORK

**Neural Scaling Law**  The classical neural scaling law typically examines the relationship between neural networks' performance and key factors such as the number of parameters $N$, training data size $D$, and the training costs $C$. Kaplan et al. (2020) showed that the test loss $L$ of a Transformer model follows a power-law relationship with key factors $N$, $D$ and $C$, when the other two factors are not constrained:

$$L = L_\infty + (\frac{x_0}{x})^{\alpha_x} \tag{1}$$

where $x$ can be $N$, $D$ or $C$, depending on which factor is being varied. $L_\infty$, $x_0$ and $\alpha_x$ are parameters that need to be determined by fitting. The authors also pointed out that, when the computation budget $C$ is fixed, scaling the model size is more important than scaling the training data. Hoffmann et al. (2022) delivered a similar power-law relationship, but they argued that $N$ and $D$ should be scaled up proportionally to achieve optimal performance under a limited computation budget. While the scaling law has been studied in various domains (Zhai et al., 2022; Aghajanyan et al., 2023; Fang et al., 2024), there remains a gap in research regarding the scaling effects on generalization capabilities across downstream tasks of varying difficulty and complexity. To the best of our knowledge, we are the first to systematically investigate the scaling effects of parameters $N$ and training data size $D$ on the generalization capabilities across downstream tasks of varying complexity.

**LLM Medical Evaluation**   Existing evaluations of LLM medical capabilities typically utilize the medical QA datasets, where the questions are sourced from medical exams (Jin et al., 2021; Pal et al., 2022; Cai et al., 2024), medical literature (Jin et al., 2019), and consumer health questions (Ben Abacha et al., 2017; Singhal et al., 2023). Other evaluation benchmarks (Zhu et al., 2023) also utilize traditional NLP tasks such as NER, relation extraction to evaluate the medical capabilities of LLMs. Recent studies (Nori et al., 2023; Singhal et al., 2023) indicate that several LLMs perform notably on these medical benchmarks. For example, GPT-4 achieves an accuracy of 90.2 on the medical exam benchmark MedQA, approaching the experts' performance on this benchmark. In this study, we create reasoning-complex tasks by reformulating the MedQA benchmark to investigate the scaling effects of model parameters and training data size on medical reasoning capabilities.

## 3   METHODOLOGY

### 3.1   PROBLEM FORMULATION

We first formalize the studied problem in this section. For an LLM with $N$ non-embedding parameters, trained on a dataset consisting of $D$ tokens, we aim to study the relationship between these two key factors and the model's performance on a series of $K$ medical downstream tasks with different levels of reasoning complexity:

$$P_i = f_i(N, D) \quad 1 \leq i \leq K \tag{2}$$

where $P_i$ denotes the performance of $i^{th}$ downstream task and $f_i$ characterizes the scaling effects of $N$ and $D$ on the performance, which is the primary focus of this study. Considering the limited research conditions[1], we primarily focus on how performance changes by varying the training data size while keeping the model parameter size fixed, and study the impact of model size by comparing the effects of data scaling across different model sizes. Inspired by the scaling law (Equation 1), we hypothesize that a power-law relationship also exists between the training data size $D$ and the model's downstream task performance $P_i$ when $D$ is sufficiently large[2]:

$$P_i = P_i^{MAX} - (\frac{D_0}{D})^{\alpha_{N,i}} \quad \text{when } D \text{ is sufficiently large} \tag{3}$$

where $P_i^{MAX}$ denotes the performance upper bound of the $i^{th}$ task, $D_0$ and $\alpha_{N,i}$ are parameters to be determined by curve fitting. The exponent term $\alpha_{N,i}$, which varies across different parameter sizes and tasks, determines the rate at which the error $P_i^{MAX} - P_i$ decays as the training data size $D$ increases. For instance, every $x$-fold increase in $D$ leads to a reduction of error to $x^{-\alpha_{N,i}}$ of its original value. We refer to Equation 3 as the *difficulty-dependent scaling law*, as we find that the error reduction rate varies largely across tasks with different levels of reasoning difficulty (complexity).

### 3.2   FACTORS OF REASONING COMPLEXITY

Though solving problems in current medical benchmarks also requires reasoning capabilities to some extent, handling real-world clinical tasks typically demands significantly more complex rea-

---

[1]Given that different LLM series trained on datasets with varying sizes, and within the same series, typically only 2-3 model sizes are open-sourced, studying the scaling effects of model parameters on performance under fixed training data size is extremely challenging.

[2]The proposed empirical power-law may hold only when $D$ is sufficiently large given the lower bound of downstream performance.

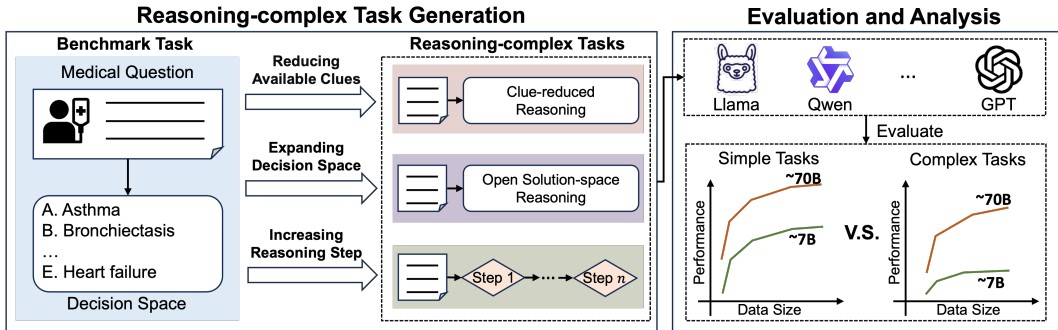

Figure 2: Overview of our study. Left: the generation process of reasoning-complex tasks in the proposed MedResEval benchmark. Right: evaluation and analysis of the scaling effects of LLMs on the constructed benchmark.

soning. In our study, we primarily focus on three key factors that cause the difference between existing benchmarks and real clinical tasks regarding reasoning complexity:

- **Available clues**: Current medical benchmark questions often provide extra clues within the context, which can aid LLMs in predicting the correct answer with tricks. For example, in multiple-choice questions (MCQs), there is only one correct answer among the options, allowing LLMs to reach the correct answer through the process of elimination rather than deriving it directly. However, real-world clinical scenarios typically offer less clues (e.g., no option provided), leading to more challenging reasoning.

- **Decision space**: The decision space of existing evaluation tasks is quite restricted. For example, the decision space of MCQs contains limited options (typically 4-6 options). In contrast, the decision spaces for real-world tasks are typically much larger. For instance, in clinical diagnosis, there may be hundreds of potential candidate diseases, requiring more complex reasoning to reach the correct diagnosis within a broader decision space.

- **Reasoning steps**: Solving problems in existing benchmarks often requires a single step of decision-making (e.g., choosing an option, making a judgement). In contrast, real-world tasks typically involve a more extended decision-making process. For instance, when managing a patient's care, a doctor first establishes a diagnosis and then prescribes treatment based on the diagnosis made in the previous step.

### 3.3 MEDICAL REASONING CAPABILITIES EVALUATION BENCHMARK

Studying the scaling effects of LLMs in the medical domain requires further evaluating them on tasks with reasoning complexity approaching the level of real-world clinical scenarios. Although some existing medical datasets involve real clinical tasks, variations in the underlying medical knowledge across these benchmarks may introduce bias in performance comparisons across tasks. To address this, we developed a new benchmark, MedResEval, by reformulating the well-known medical benchmark MedQA (Jin et al., 2021), which contains multiple-choice questions sourced from United States Medical Licensing Examination (USMLE). Tasks in MedResEval are designed to require more complex reasoning and decision-making capabilities related to the key factors introduced above, leading to more accurate reflection of the reasoning complexity involved in real medical tasks. The generation process of this benchmark and our evaluation process are illustrated in Figure 2.

**Evaluation Tasks**  Given the key factors of reasoning complexity, we first devise three types of reformulated tasks that focus on reducing available clues, expanding decision space, and increasing reasoning steps, respectively:

- **Reducing Available Clues**: We consider reducing the extra available clues provided in the MCQ options by combining the question part of each MCQ sample with one of its options, resulting in a statement verification task. To verify the statements, LLMs must rely solely

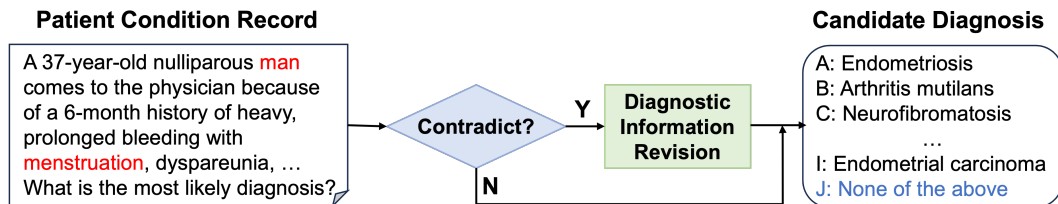

Figure 3: The proposed diagnosis-simulation task requires LLMs to first identify and revise the contradiction in the patient condition record, followed by selecting the correct diagnosis from a larger set of candidates.

on the information in the question and the content of statements, without the additional clues provided by comparing choices.

- **Expanding Decision Space**: Inspired by Rajpurkar et al. (2018), we generate unanswerable questions to expand the decision space of original MCQ questions. Specifically, we randomly replace the correct answer of each MCQ question with a wrong option, resulting in a corresponding unanswerable question. Next, we ask LLMs to determine whether the correct answer is included among the options of a question extracted from either the original dataset or the reformulated dataset. To make the judgment, LLMs must consider possible answers beyond the given options, leading to a more open decision space.

- **Increasing Reasoning Steps**: To increase the reasoning steps, we design a simple two-hop reasoning task where LLMs first verify the correctness of an answer to the given MCQ, and then select the correct option if the provided answer is incorrect. Solving this task requires LLMs to make consistent decisions across steps, emphasizing their reasoning consistency throughout the increased reasoning steps.

Although the above tasks evaluate LLMs' medical reasoning capabilities from three separate aspects, real medical tasks often require integrating multiple reasoning skills simultaneously. Moreover, considering that real-world medical context may contain noise and contradictions, the ability to identify and revise contradictory information is crucial for real-world clinical tasks, such as diagnosis. Therefore, we design a more complex **diagnosis-simulation** task to evaluate LLMs' ability to apply multiple reasoning skills simultaneously. Figure 3 presents an example of this task. Specifically, we perturb the key diagnostic information (gender, examination results, symptoms) in the original diagnosis-type MCQs. Next, we have three experienced doctors annotate whether a contradiction exists in the perturbed questions. For each perturbed question, LLMs should first determine if a contradiction exists. If so, they must revise it by choosing the appropriate revision option before providing the final diagnosis; otherwise, they directly answer the original question. We further expand the decision space of this task by adding more candidate diseases and randomly removing the correct diagnosis from the options, requiring LLMs to select "None of the above" in such cases.

**Task-wise Performance Alignment**  To study the scaling effects of LLMs on medical reasoning capabilities, it is crucial to compare their performance across tasks of varying reasoning complexity. However, due to the differing formats of each task, performance metrics must be further aligned for meaningful comparison. For example, a 50% accuracy on a 5-choice MCQ task is relatively high, while the same accuracy is close to random guessing in a statement verification task with balanced labels. Therefore, we normalize the performance of downstream tasks as follows:

$$P_i^{norm} \triangleq \frac{P_i - P_i^{rand}}{P_i^{MAX} - P_i^{rand}} \tag{4}$$

Here, $P_i^{rand}$ is the random guessing performance on the $i^{th}$ task. Note that $P_i^{norm} = 1$ indicates the upper limit of performance, $P_i^{norm} = 0$ refers to the random guessing performance, and $P_i^{norm} < 0$ indicates performance worse than random guessing.

## 4 EXPERIMENT SETUP

**Datasets**   We primarily conduct experiments on both the original MedQA benchmark and the proposed MedResEval benchmark to compare the scaling effects of LLMs on tasks that require varying levels of medical reasoning capabilities. However, given that the differences in results between MedResEval and MedQA could also be due to simple generalization effects, it is important to introduce an additional baseline task to eliminate the impact of generalization. On the other hand, apart from reasoning capabilities, mastering the diverse semantic representations of the same medical concept is also crucial for addressing real-world medical tasks. Therefore, we construct another **semantic-varied** task by replacing the medical terms in the original MedQA dataset with corresponding synonyms. By introducing this baseline task, we can eliminate the effects of simple generalization and further explore the differences in scaling effects across various dimensions of medical capabilities, including semantic understanding and reasoning capabilities. More details of dataset generation, including the statistics and examples of tasks, are provided in Appendix A.

**Evaluation Setting**   For evaluation, we employ the 5-shot Chain-of-Thought (CoT) setting (Wei et al., 2022) to maximize the evaluation of LLMs' potential in the medical domain. Following (Nori et al., 2023), we use GPT-4o to generate CoTs that are consistent with the ground truth labels. Additionally, we apply the self-consistency approach (Wang et al., 2022) to enhance performance and ensure evaluation stability. We used handcrafted regular expressions to parse the answers from the LLM responses and found that they successfully extracted the answers most of the time. We select accuracy as the performance metrics, and conduct the normalization mentioned in Equation 4. More details of evaluation (e.g., prompt formats, inference setting) are provided in Appendix B.

**Evaluated Models**   We conduct most of the experiments and analysis with LLMs from the Llama (Touvron et al., 2023a;b; Dubey et al., 2024) and Qwen families (Bai et al., 2023; Yang et al., 2024), since these two model families span a broader range of parameter sizes and pretraining data sizes than other LLM series, and are widely used for developing downstream applications. Specifically, we evaluate 12 LLMs across three generations and two model sizes ($\sim$7B, $\sim$70B)[3]. We also evaluated several instruction-tuned LLMs (Qwen2-Instruct, Qwen2.5-Instruct) to study the effect of instructing tuning on the scaling effects. Finally, we evaluated GPT-4o and GPT-4o-mini (OpenAI, 2024)) for reference. Statistics of the evaluated LLMs are listed in Table 3 of Appendix C.

## 5 RESULTS

### 5.1 MAIN ANALYSIS

**Qualitative Analysis**   We begin by comparing the scaling effects of LLMs on medical tasks with varying levels of reasoning complexity. Specifically, we split the evaluation tasks into two groups: (1) Reasoning-simple tasks, including the original MedQA task and the semantic-varied task; (2) Reasoning-complex tasks, including the reducing-available-clues task, the expanding-decision-space task, and the increasing-reasoning-steps task[4]. Figure 4 presents the results of this analysis. We observed that **for simple tasks, scaling model size and training data size by the similar factor yields similar performance improvement**, and those smaller models trained on much larger datasets have matched or even outperformed larger models trained on smaller datasets. For example, Llama3-8B ($\sim$7B, 15T) performs comparably to Llama-65B ($\sim$70B, 1.4T) on simple tasks (34.4% vs. 31.6%), despite being 8x smaller but trained on 10x more data. However, **for tasks that require more complex reasoning, scaling model size by the similar factor results in significantly greater gains compared to scaling data by the similar factor**. The experimental results show that Llama models with $\sim$7B parameters consistently underperform compared to Llama-65B, even when trained on 10x more data. A similar trend is also observed in the Qwen model family, with the latest Qwen2.5-7B underperforming Qwen-72B by 10% on complex tasks. In general, **as tasks require more complex reasoning, the effect of scaling model size becomes much more pronounced than scaling data size**.

---

[3]We primarily focus on 7B and 70B LLMs since open-source LLMs of these scales are relatively abundant and have been widely applied in various downstream tasks.

[4]Considering that the diagnosis-simulation task is derived from part of the original benchmark (diagnosis questions), we do not directly compare its performance with other tasks.

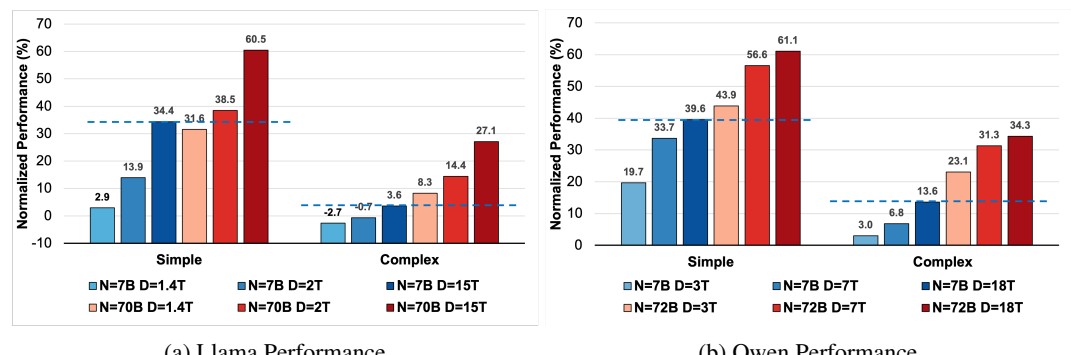

(a) Llama Performance           (b) Qwen Performance

Figure 4: Performance (normalized) of LLMs with varying model sizes and training data sizes on different levels of reasoning complexity in MedResEval. Blue bars: ∼7B LLMs; Red bars: ∼70B LLMs. The depth of color represents the amount of training data.

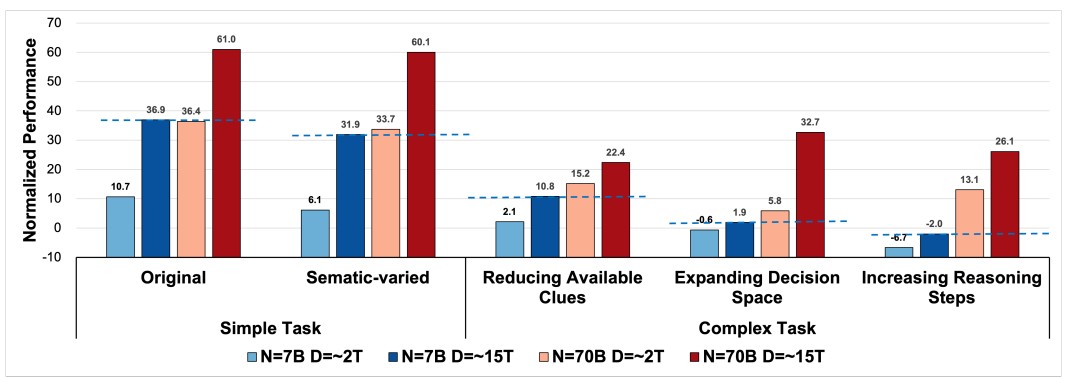

Figure 5: Performance (normalized) of Llama models with varying model sizes and training data sizes on different medical tasks in MedResEval.

We further investigate the fine-grained scaling effects across various tasks in the proposed MedResEval. Taking the Llama series as an example (the complete evaluation results are provided in Appendix D), we categorize the models into four groups based on the number of training tokens (less (1.4T, 2T), more (15T)) and the parameter size (less (∼7B), more (∼70B)), and average the performance of models within each group. Figure 5 presents the performance of each group on different downstream medical tasks. On the original benchmark task, we observe that the latest 7B model trained on 15T tokens has surpassed 70B models trained on ∼2T tokens. The results on the semantic-varied task are very similar to those on the original task, suggesting that current LLMs have mastered the semantic representations of medical concepts quite well. However, we found that the 70B models consistently outperform the 7B models across all three reasoning-complex tasks, particularly in the increasing-reasoning-steps task. Such phenomenon indicates that **simply scaling up the training data size of small-scale LLMs cannot effectively enhance their capabilities to handle more reasoning steps**. We also observe that on the expanding-decision-space task, scaling up the training data leads to only a 5% improvement for 7B models, whereas it results in a ∼25% improvement for 70B models. These results suggest that enhancing reasoning capabilities in a more open decision space requires simultaneous scaling of both the training data size and the model size, as increasing the training data size alone is insufficient.

**Quantitative Analysis**    To better understand the scaling effects on medical tasks with varying levels of reasoning complexity, we further conduct a quantitative analysis by plotting the evaluation results against the number of training tokens and the *normalized error rate* $1 - P_i^{norm}$ (see Figure 6). We observe an approximate power-law relationship between the training data size and the normalized error rate, which confirms our hypothesis in Section 3.1 (Equation 3). We further conduct curve fitting using the evaluated performance and the reported training data sizes of LLMs with

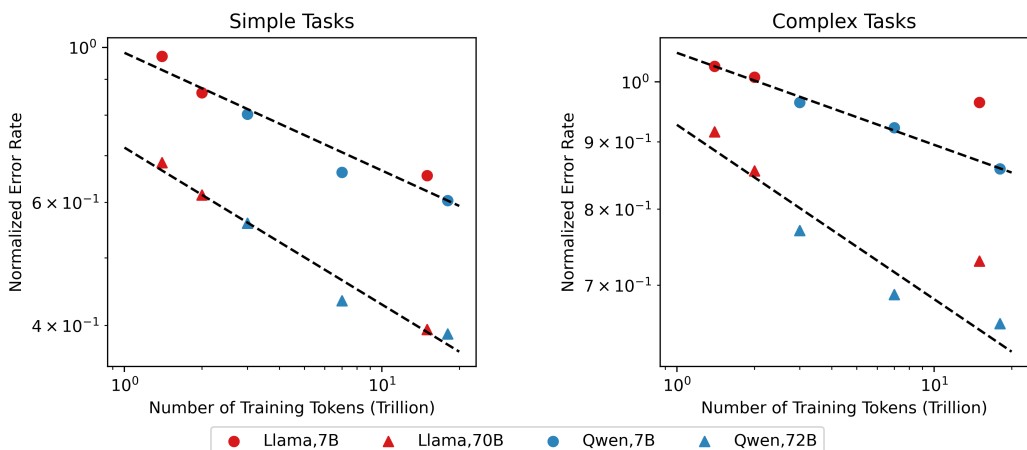

Figure 6: The performance of LLMs with fixed model sizes varies predictably with training data size $D$, with curves (dashed lines) fitted based on the *difficulty-dependent scaling law* (Equation 3).

| Task Type | #Model Params | Fitting Formula | $R^2$ |
|---|---|---|---|
| Simple | $\sim$7B | $P = 1 - 0.982D^{-0.168}$ | 0.944 |
|  | $\sim$70B | $P = 1 - 0.719D^{-0.224}$ | 0.978 |
| Complex | $\sim$7B | $P = 1 - 1.052D^{-0.067}$ | 0.991 |
|  | $\sim$70B | $P = 1 - 0.927D^{-0.132}$ | 0.919 |

Table 1: Formulas of the fitting curves presented in Figure 6 with $R^2$ scores (some outliers are discarded in calculating $R^2$).

varying model sizes across tasks of different reasoning complexity[5]. The details of curve fitting is provided in Appendix E, while the fitting results are presented in Table 1. Overall, the fitting curves achieve relatively high $R^2$ scores, which quantitatively supports the validity of our proposed empirical formula. Only a few models (e.g., Llama3) deviate significantly from the predicted curves, possibly due to other factors such as training data quality. Moreover, the fitting results indicate that **the profit of scaling training data are greater for models with larger sizes, especially for tasks that rely on complex reasoning**. Based on the fitting equations, a 7B model trained on $\sim$54.7T tokens can achieve the same performance on simple tasks as a 70B model trained on 5T tokens; however, to match the performance of that 70B model on complex tasks, the 7B model would need to be trained on $\sim$157T tokens, which is 10 times the number of tokens used for training Llama3. Therefore, considering the difficulty of collecting high-quality training data, **the approach of scaling up the training data size to further improve the performance of small-scale LLMs on real-world medical tasks, which requires more complex reasoning, is unsustainable**.

## 5.2 Case Study on Diagnosis-simulation Task

We also conduct a case study on the proposed diagnosis-simulation task that requires revising possible contradictions in the patient condition and then making the correct diagnosis. The performance of LLMs on this task is presented in Figure 7. Generally, larger LLMs consistently outperform smaller LLMs on this task, which exhibits similar trends compared to reasoning-complex tasks. Moreover, we observe that the performance of 70B LLMs improves more rapidly with increased training data compared to 7B LLMs: for the 7B Llama models, a 10x increase in data resulted in approximately an 8% performance improvement, whereas the 70B Llama model saw about a 20% improvement. Qwen LLMs did not exhibit consistent performance gains with increased data, which may be attributed to differences in training data distribution. Nevertheless, we also observe that increasing the training data size from 7T to 18T leads to a 7% performance improvement for the 70B Qwen LLMs, while the performance of the 7B LLMs remains unchanged. Such phenomenon sug-

---

[5]Some outliers were not considered in the fitting.

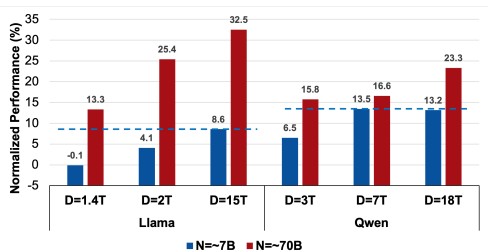

| Model | Simple | | | Complex | | |
|---|---|---|---|---|---|---|
| | Base | IT | Δ | Base | IT | Δ |
| Qwen2-7B | 33.7 | 31.9 | -1.8 | 6.8 | 18.6 | 11.7 |
| Qwen2-72B | 56.6 | 60.5 | 3.9 | 31.3 | 46.5 | 15.1 |
| Qwen2.5-7B | 39.6 | 42.7 | 3.1 | 13.7 | 23.3 | 9.6 |
| Qwen2.5-72B | 61.1 | 65.5 | 4.4 | 34.2 | 49.9 | 15.7 |
| GPT-4o-mini | - | 67.0 | - | - | 43.5 | - |
| GPT-4o | - | **82.1** | - | - | **69.3** | - |

Figure 7: Performance of LLMs with varying model sizes and training data sizes on the diagnosis-simulation task.

Table 2: Impact of instruction tuning on the downstream performance of LLMs with varied model sizes and training data sizes.

gests that further increasing the training data size of 7B LLMs may not lead to better performance on tasks involving complex reasoning skills.

### 5.3 IMPACT OF INSTRUCTION TUNING

Finally, we also investigate the impact of instruction tuning on downstream performance of LLMs with varied model sizes and training data sizes. Specifically, we evaluate four instruction-tuned LLMs with varying model sizes and training data sizes, and compare them to their respective base models. We also evaluate two flagship commercial LLMs (GPT-4o-mini and GPT-4o) for reference. The experimental results are presented in Table 2, with detailed performance provided in Appendix D. The experimental results demonstrate the effectiveness of instruction tuning, as it largely improves performance on complex tasks by $> 10\%$. However, **instruction tuning does not reduce the performance gap between models with different parameter scales**. In fact, for complex tasks, the performance gap between 7B and 70B models widened by $\sim 5\%$ after instruction tuning. This further confirms the generalizability of our conclusions regarding parameter and training data scales. Finally, we observe that although GPT-4o-mini outperforms all of the evaluated LLMs on simple tasks, it falls behind Qwen2.5-72B on complex and real-world simulation tasks. In contrast, GPT-4o significantly outperforms all of the evaluated LLMs across tasks of varying difficulty levels. This further underscores the importance of scaling model size for addressing complex real-world problems. It is worth noting that a large gap (13%) still exists between GPT-4o's performance on simple and complex tasks, indicating that the current ability of LLMs to generalize medical knowledge across different tasks requires further improvement.

## 6 CONCLUSION

Solving real-world clinical problems require LLMs to master advanced medical reasoning capabilities. In this paper, we revisiting the scaling effects of LLMs on medical reasoning capabilities. We focus on three key factors that affect the reasoning complexity, and devise a specific task that simulates a problem from a real medical scenarios. Based on the evaluation of 18 LLMs with varying model sizes and training data amounts, we find that though some latest 7B LLMs have achieved comparable or even superior performance than elder 70B LLMs on existing benchmarks, they perform consistently worse than the latter on tasks involving more complex reasoning. Furthermore, our quantitative analysis demonstrate that scaling the model size always lead to a more pronounced performance improvement than scaling data size, especially on reasoning-complex tasks. Further investigation indicates that though instruction-tuning does improve LLMs' reasoning capabilities to some extent, the scaling effects of model size are still notable. Our study findings indicate that while both data and parameter scales enhance LLM performance, greater emphasis must be placed on model parameter scales, particularly for difficult reasoning tasks. LLMs with sufficiently large-scale model size are essential for effectively addressing difficult reasoning tasks, such as some real-world medical problems encountered.

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

## A  DETAILS OF BENCHMARK CONSTRUCTION

We construct MedResEval by reformulating the original MedQA dataset, as introduced in preceding sections. MedQA is a large-scale medical exam dataset, containing medical exams sourced from three different regions. In this study, we use the questions in the US subset, which are collected from the United States Medical Licensing Examination (USMLE). Questions in MedQA are MCQs with five options, as shown in the top of Figure 8. Typically, a MedQA question consist of introduction of

a patient's condition, together with a corresponding question part regarding the diagnosis, treatment, or other aspect of the patient. For task reformulation, we first use regular expressions to identify and parse the question part to determine the type of question. Furthermore, we generate three types of reasoning-complex questions as follows:

- reducing-available-clues questions: We combine the extracted question part of the MCQ with a correct answer and a randomly selected incorrect answer, resulting in two statements. To increase the reasoning difficulty, we further derive two additional statements by adding negation words. Finally, we concatenate these statements with the description of the patient's condition.

- expanding-decision-space questions: We replace the correct option in the original MCQ with a distractor generated by GPT-4 to create unanswerable questions. To ensure the accuracy of the generated questions, we had two experienced doctors on our team review the reformulated questions.

- increasing-reasoning-steps questions: We generate this type of question by providing a possible answer and instructing LLMs to verify the given answer, correcting it if it is incorrect. For example, we might ask, "Alice chose answer [given answer]. Please verify if this is correct, and if not, provide the correct answer." In our implementation, we generate two questions based on the original MCQ by providing a correct answer and a randomly selected incorrect answer.

Examples of these three types of are provided in the bottom of Figure 8. Based on a total of 800 parsable questions sourced from the MedQA test set, we generate 3,200 questions for the reducing-available-clues type, and 1,600 questions each for the expanding-decision-space type and increasing-reasoning-steps type.

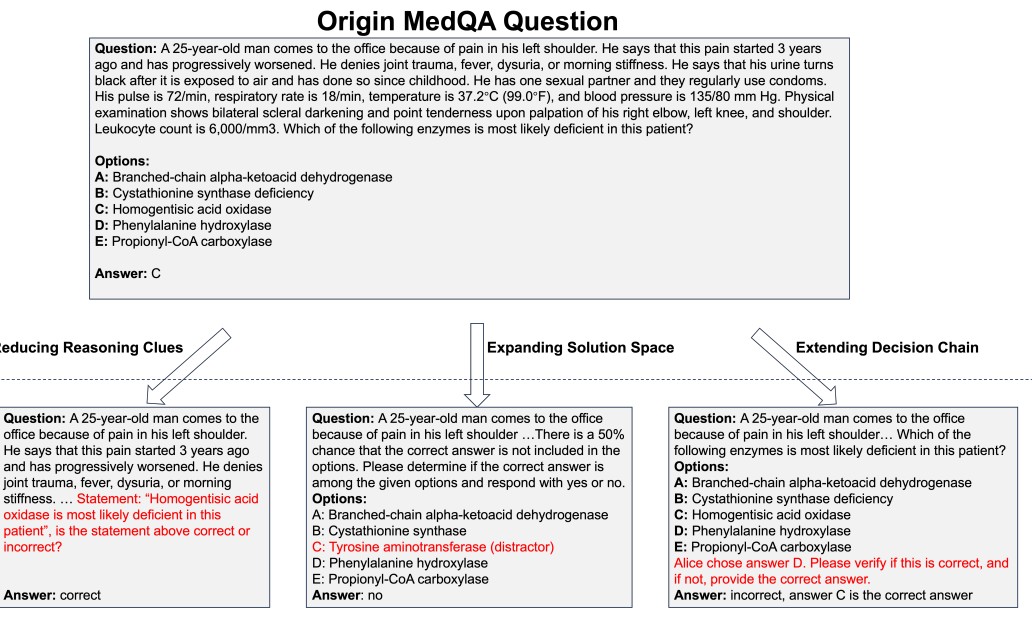

Figure 8: An example of reformulating questions in the MedQA dataset.

We also generate diagnosis-simulation questions based on the MedQA dataset. Figure 10 illustrates an example of this type of question. For the diagnosis-simulation task, given that the number of diagnosis-type questions in the MedQA test set is relatively low (fewer than 100 questions), we collect all diagnosis-related questions from the entire MedQA dataset, resulting in 900 questions. We then use regular expressions to identify and perturb key information, such as gender, examination results, and symptoms. Next, we expand the original option set by randomly sampling four negative

diseases and adding a "none of the above" option. We shuffle the sequence of options and randomly replace the correct answer with a distractor disease in 10% of the questions to balance the selection probability of options and avoid potential positional bias. Finally, two doctors review the perturbed questions to determine whether any contradictions exist; if not, they assess whether a correct answer is present among the options. We also generate three revision options, with two incorrect options that replace the correct information with the perturbed details, and one option that contains the correct revision. As a result, a total of 1,989 questions are generated through this process.

We also construct a semantic-varied task by replacing the medical terms in the questions of the MedQA benchmark. Specifically, we use a medical NER tool MedCAT (Kraljevic et al., 2021) to identify the medical terms presented in the questions. Then we search in the UMLS database to retrieve the synonyms of these medical terms. Finally, we randomly replace these identified medical terms with corresponding synonyms to varied the original questions in the semantic-level.

**Question**: A 28-year-old man at 30 weeks gestation is rushed to the emergency room with the sudden onset of vaginal bleeding accompanied by intense abdominopelvic pain and uterine contractions. ... Which of the following is the most likely diagnosis in this patient?
**Options:**
A: Vasa previa
B: Placenta abruption
C: Eczematous dermatitis
D: Uterine rupture
E: Adrenocortical carcinoma
F: Miscarriage
G: Attention deficit hyperactivity disorder
H: Placenta previa
I: Extranodal marginal zone lymphoma
J: None of the above
Is there any contradiction in the patient's gender, examination items, symptoms, or other key diagnostic information? If there is, please select the correct revision process from the following options and answer the revised question; if not, please answer the original question.
**Revision options**:
1: Results of temperature examination: 37.0°C -> 36.5°C
2: Symptoms: sudden onset of vaginal bleeding -> no sudden onset of vaginal bleeding
3: Gender: Male -> Female

Figure 9: An example of the generated diagnosis-simulation question.

# B  DETAILS OF EVALUATION SETTING

We illustrate the prompt format used for evaluation, which combines five-shot Chain of Thought (CoT) prompting with self-consistency reasoning to maximize the performance of large language models (LLMs) on challenging reasoning tasks. Specifically, we concatenate five demonstrative question examples with the test question, where each example consists of a question, a CoT, and the corresponding answer. Using this constructed input sequence, we prompt the evaluated LLM five times to obtain five different reasoning chains. We set the temperature to 0.8 and top_p to 0.8 to encourage more diverse responses. Finally, we use regular expressions to extract answers from the responses and conduct a majority vote to determine the final prediction.

For the generation of CoT, we follow (Nori et al., 2023) by prompting GPT-4o to generate CoT based on the demonstrative question using the following template: *[Question]. Please provide a complete chain-of-thought to answer the given question.* We prompt the GPT-4o several times for each questions, only keeping the CoTs that are consistent with the ground-truth label.

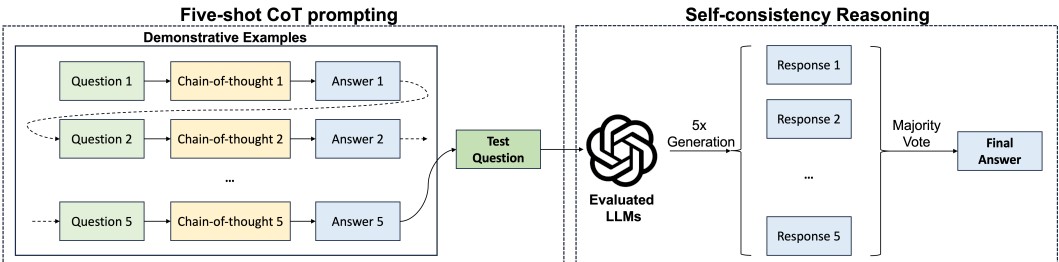

Figure 10: An example of the generated diagnosis-simulation question.

# C  STATISTICS OF EVALUATED LLMS

We list the number of parameters and training tokens in Table 3, regarding the LLMs studied in this paper.

| Model | #Parameter (B) | #Training Tokens (T) |
|---|---|---|
| Llama | 7/65 | 1.4 |
| Llama2 | 7/70 | 2 |
| Llama3 | 8/70 | 15 |
| Qwen | 7/72 | 3 |
| Qwen2 | 7/72 | 7 |
| Qwen2.5 | 7/72 | 18 |

Table 3: Statistics of studied LLMs in this paper.

## D    DETAILED EVALUATION RESULTS ACROSS TASKS

| Model | Origin | SemVar | ReduAvaClu | ExpDeciSpac | IncReasStep | Diagnosis |
|---|---|---|---|---|---|---|
| llama-7B | 1.1 | 4.7 | 2.1 | 0.5 | -11.1 | -6.4 |
| llama2-7B | 20.3 | 7.5 | 2.2 | -2.1 | -2.4 | -3.1 |
| llama3-8B | 36.9 | 31.9 | 10.8 | 1.9 | -2.7 | 6.4 |
| Qwen-7B | 21.2 | 18.2 | 13.1 | -1.4 | -1.0 | 4.1 |
| Qwen2-7B | 34.0 | 33.5 | 14.2 | 6.0 | 3.0 | 11.9 |
| Qwen2.5-7B | 42.0 | 37.3 | 17.5 | 10.6 | 14.3 | 17.5 |
| llama-65B | 34.1 | 29.1 | 13.3 | 4.9 | 6.7 | 5.1 |
| llama2-70B | 38.7 | 38.4 | 17.1 | 7.0 | 21.3 | 13.3 |
| llama3-70B | 61.0 | 60.1 | 26.5 | 32.5 | 26.1 | 24.1 |
| Qwen-72B | 45.8 | 42.1 | 26.4 | 11.1 | 31.4 | 22.0 |
| Qwen2-72B | 56.6 | 56.6 | 36.1 | 21.8 | 35.6 | 23.7 |
| Qwen2.5-72B | 63.5 | 58.6 | 36.4 | 22.9 | 44.5 | 28.8 |
| Qwen2-7B-Instruct | 33.0 | 30.8 | 18.8 | 11.1 | 25.7 | 12.2 |
| Qwen2-72B-Instruct | 61.6 | 59.4 | 48.2 | 30.4 | 60.9 | 30.7 |
| Qwen2.5-7B-Instruct | 44.7 | 40.7 | 17.7 | 19.1 | 32.8 | 17.4 |
| Qwen2.5-72B-Instruct | 67.9 | 63.1 | 54.9 | 30.7 | 64.0 | 44.6 |
| GPT-4o-mini | 69.2 | 64.8 | 52.1 | 18.0 | 60.6 | 31.9 |
| GPT-4o | 83.2 | 81.0 | 75.3 | 50.7 | 81.9 | 47.7 |

Table 4: Full evaluation results of LLMs on the proposed MedResEval benchmark.   Sem-Var: semantic-varied baseline task; ReduAvaClu: reducing-available-clues task; ExpDeciS-pac: expanding-decision-space task; IncReasStep: increasing-reasoning-steps task; Diagnosis: diagnosis-simulation task.

## E    DETAILS OF CURVE FITTING

We conduct linear regression between the normalized log error rate $\log(1 - P)$ and the log training data size $\log(D)$ using the following formula:

$$\log(1 - P) = -\alpha_{N,i} \log(D) + A \tag{5}$$

where $A = \log(D_0)$. Note that this formula is equivalent to the Equation 3 presented in preceding sections. For simple tasks, we use all the evaluation results in the curve fitting; for complex tasks, we observe that the results of Llama3 behave as outliers in the fitting, while the fitting results based on the rest LLMs achieve relative high $R^2$ scores. Therefore, we do not consider the Llama3 results in curve fitting.

