# OpenReview forum: "Revisiting the Scaling Effects of LLMs on Medical Reasoning Capabilities"
_ICLR.cc/2025/Conference — Submitted to ICLR 2025_

### Official Review · Reviewer_Xehd · 2024-10-26

**Soundness:** 2
**Presentation:** 2
**Contribution:** 1
**Rating:** 3
**Confidence:** 4

**Summary:**

This paper identifies the lack of a robust dataset to benchmark the reasoning capabilities of Large Language Models (LLMs) in complex medical scenarios. To address this gap, the authors adapt the MedQA dataset, creating a new benchmark called MedResEval with three key improvements: limited clues, a broader decision space, and additional reasoning steps. The authors then benchmark multiple open-source LLMs on this dataset and propose scaling laws that relate performance to training data size.

**Strengths:**

- The paper addresses an important issue in evaluating LLM reasoning in the medical field.
- The experiments are conducted on a wide range of LLMs.

**Weaknesses:**

- The novelty of the proposed dataset fall short when compare  to existing datasets:
	- The authors argue that MCQs provide too many clues and a limited decision space. However, the modified dataset they propose still contains only MCQs, despite the existence of medical question-answering datasets without MCQs [1].
	- The authors propose benchmarking the multistep reasoning abilities of LLMs by artificially adding a reasoning step to the MedQA dataset. However, datasets specifically designed to assess this ability already exist [2,3], making the novelty of the authors' benchmark relatively limited in comparison.

- The benchmark proposed by the authors utilizes "Chain of Thought" prompting, with demonstrations generated by GPT-4. This approach makes the benchmark dependent on the performance of a third-party, closed-source model, and it diverges from realistic medical scenarios, as sensitive medical data cannot be processed by GPT-4 due to ethical concerns.


- The experimental details are incomplete, particularly the absence of the specific prompts used. This omission makes it challenging to have confidence in the results and to reproduce them, as the performance of each LLM can vary significantly depending on the prompt used.

- The paper lacks a contribution section, which makes it difficult to discern the specific claims and contributions being presented.

- The experiments lack reported margins of error, making it difficult to evaluate the significance of the presented results.

[1] (2018) emrQA: A Large Corpus for Question Answering on Electronic Medical Records

[2] (2018) Can a Suit of Armor Conduct Electricity? A New Dataset for Open Book Question Answering

[3] (2022) MuSiQue: Multihop Questions via Single-hop Question Composition

**Questions:**

- Is it possible that the dataset being enhanced is part of the training set for some of the LLMs used in this study? Given that MedQA was released in 2021 and the Llama-3 models’ training data has a cutoff in December 2023 for example, it seems likely that this dataset or its metadata could have been included in the training data. This issue is critical to consider for two reasons:
	- It may bias the proposed benchmark, as some models might have been trained on this dataset while others have not. Additionally, it risks transforming the benchmark into a test of memorization rather than reasoning ability.
	- It diverges from the real-world clinical conditions the authors aim to simulate,  particularly in comparison to other benchmarks that ensure they are not included in the training sets of LLMs [1,2].

- In Section 4, the authors propose adding an additional baseline to address the "simple generalization effect." Could the authors clarify what specific effects they are referring to here and explain how the added baseline mitigates these effects? Citing relevant literature to support this would be appreciated.

[1] (2018) emrQA: A Large Corpus for Question Answering on Electronic Medical Records

[2] (2022) DrugEHRQA: A Question Answering Dataset on Structured and Unstructured Electronic Health Records For Medicine Related Queries

---

> ### Author Response · Authors · 2024-12-04
> **Response to Reviewer Xehd (1/3)**
>
> We are sincerely grateful for your detailed and constructive feedback. Below, we provide our responses to each of the concerns you raised.
>
> 1. **Novelty of Our Work**: Thank you very much for your meticulous review. It is important to emphasize that the core contribution of our work is not the construction of a benchmark for evaluating LLM medical reasoning capabilities, but rather the systematic analysis of the scaling effects of LLMs on tasks with varying reasoning difficulties. In fact, prior to conducting this study, we observed that some of the latest small-scale LLMs (e.g., LLaMA3-8B) achieved comparable or even superior performance to larger-scale LLMs across various benchmarks, including famous medical benchmarks like MedQA. The remarkable performance of these small-scale LLMs has led some researchers to believe that small-scale LLMs trained on larger-scale datasets can rival the performance of larger-scale LLMs. The primary goal of our study is to empirically analyze the effects of training data size $N$ and model parameter size $D$ on LLMs’ reasoning abilities by comparing the performance of a series of LLMs on tasks with varying reasoning difficulty. Our evaluation results reveal that while the latest small-scale LLMs achieve performance comparable to larger-scale LLMs on simpler tasks, their performance falls significantly behind on more challenging reasoning tasks. This indicates that for complex reasoning tasks, merely scaling up the training data size is insufficient; scaling up the model parameter size is also essential. The conclusions drawn from this study provide valuable guidance for the future development and application of LLMs in reasoning tasks.
>
>    + **The Modified Dataset Still Contains Only MCQs**: In our work, we considered that existing MCQs often provide extra reasoning clues, have a relatively limited decision space, and require fewer reasoning steps. To address this, we modified MCQs from existing medical benchmarks as follows: (1) For reducing reasoning clues, we combined the questions and options of existing MCQs into statement verification tasks, preventing LLMs from leveraging comparisons between options to gain additional reasoning clues; (2) For expanding decision space, we replaced the correct options in MCQs with distractors, requiring LLMs to determine whether the correct answer lies in the options or not; (3) For increasing reasoning steps, we added an additional reasoning step where LLMs must first judge whether the given answer is correct for the original MCQ and, if not, provide the correct answer. While the modified questions still include some form of options (e.g., true/false, answer exists/does not exist), these modifications do enhance the reasoning complexity of the original tasks from three aspects, supporting our analysis of the scaling effects of LLMs on medical reasoning capabilities.
>    + **Why Not Use Existing Reasoning-Complex Datasets**: Thank you for your question. Actually, in our early research, we also considered directly using existing reasoning-complex medical benchmarks for analysis. However, it is crucial to note that the primary goal of this study is not to evaluate the medical reasoning capabilities of current LLMs but rather to analyze the scaling effects of LLMs on medical reasoning. For this purpose, we need to compare the performance of LLMs on both simple and complex reasoning tasks. Directly comparing results from existing medical benchmarks and reasoning-complex benchmarks would be unfair, as these benchmarks assess different medical knowledge areas, and the evaluation results would be significantly influenced by the LLMs’ level of medical knowledge mastery. Therefore, in our implementation, we modified a single medical benchmark while controlling for the medical knowledge being tested. By addressing three key factors affecting reasoning difficulty, we constructed medical benchmarks tailored to our research needs.
>
> 2. **GPT-generated CoT Issue** Thank you for your thoughtful feedback. In this study, we followed [1] to generate “Chain of Thought” (CoT) prompting using GPT-4, aiming to maximize the reasoning capabilities of LLMs. Indeed, CoT testing based on GPT-4 diverges from realistic medical scenarios. However, the primary objective of this research is not to evaluate LLMs’ reasoning performance in real-world medical contexts but to study the scaling effects of LLMs on medical reasoning capabilities. Given that inferencing using GPT-4-generated CoT represents a strong baseline method on current medical benchmarks, we primarily adopted this approach for our experiments. Based on your valuable suggestions, we plan to further validate our findings in the future using alternative settings (e.g., direct 5-shot learning).
>
>    [1] Nori H, Lee Y T, Zhang S, et al. Can generalist foundation models outcompete special-purpose tuning? case study in medicine[J]. arXiv preprint arXiv:2311.16452, 2023.

---

> > ### Author Response · Authors · 2024-12-04
> > **Response to Reviewer Xehd (2/3)**
> >
> > 3. **Experimental Details Issues**: Thank you for your constructive feedback. Due to space limitations, we have provided examples of each evaluation task and detailed prompt construction in Appendix A and Appendix B. We will further supplement the appendices with the complete prompts used for each evaluation task to enhance reproducibility.
> >
> > 4. **Contribution Section**：We greatly appreciate your constructive comments. We will add a dedicated contribution section in the revised manuscript to clearly outline the core contributions of our work.
> >
> > 5. **Significance of Results**: We are very grateful for your constructive feedback. To ensure the significance and reliability of our conclusions, we employed the self-consistency method [1] during model evaluation. Specifically, we prompted the LLMs to generate answers to the same question five times and then determined the final model prediction using a majority vote. This approach effectively enhances the performance of LLMs and contributes to the robustness of the evaluation results. Based on your kind suggestions, we also conducted two additional repeated experiments to calculate confidence intervals. Given the high computational cost of evaluating all models, we prioritized supplementary experiments on the four models from the Llama family. The results of these experiments, with 95% confidence intervals, are as follows:
> >
> >    | Task                       | llama2-7B | llama2-70B | llama3-8B | llama3-70B |
> >    | -------------------------- | --------- | ---------- | --------- | ---------- |
> >    | Original                   | 16.2±0.3  | 41.8±1.3   | 44.8±0.5  | 64.9±0.0   |
> >    | Reducing Available Clues   | 0.8±0.1   | 17.9±0.2   | 13.4±0.7  | 29.2±0.8   |
> >    | Expanding Decision Space   | 0.5±0.3   | 6.9±0.6    | 0.9±0.2   | 28.7±0.6   |
> >    | Increasing Reasoning Steps | -1.3±0.1  | 26.0±1.2   | -1.2±0.2  | 37.5±0.4   |
> >
> >    We found that the confidence intervals obtained from multiple experiments are relatively small, primarily because our test setting inherently enhances robustness. Based on your valuable feedback, we will incorporate confidence intervals into the results presented in the paper to further demonstrate the credibility of our findings.
> >
> > 6. **Risk of Benchmark Leakage**: Thank you for your thoughtful suggestion. There is indeed a possibility that MedQA might have been included in the training data of some LLMs, while our modifications to the original dataset help mitigate potential benchmark leakage. Specifically, we designed new task types to alter the reasoning structure of the original questions, making the resulting benchmark sufficiently different from the original dataset and minimizing the risk that models rely solely on memorization. On the other hand, while some existing benchmarks claim they are not included in the training sets of LLMs, it is important to note that despite agreements to prevent their use for training, the lack of transparency in current LLM training processes makes it difficult to completely eliminate this risk. In fact, all publicly available benchmarks face a similar risk of evaluation leakage. Given that MedQA is one of the most widely used datasets for evaluating LLMs’ medical capabilities and includes a large number of medical exam questions designed to evaluate human medical capabilities, we primarily based our analysis on MedQA in this study. Again, thank you for your valuable suggestions. To further eliminate the risk of benchmark leakage, we also plan to apply benchmark leakage identification methods to further enhance the fairness of the evaluation.

---

> > > ### Author Response · Authors · 2024-12-04
> > > **Response to Reviewer Xehd (3/3)**
> > >
> > > 7. **Additional Baseline Issue**: Thank you very much for your detailed review. In this work, we primarily reformulated existing medical benchmarks to create more challenging evaluation sets, aiming to analyze the scaling effects of LLMs on medical reasoning capabilities. However, the performance of LLMs on the reformulated datasets is influenced by both reasoning difficulty and generalization itself. For example, while an LLM might memorize a test sample during training, even superficial textual changes to the question could lead to degraded performance on the reformulated tasks. In our paper, we refer to this phenomenon as the “simple generalization effect.” To account for this effect in our experimental results, we introduced a baseline task (Semantic-Varied). This baseline reformulates questions by identifying and replacing medical terms in the original benchmark questions with synonyms, without significantly altering the reasoning difficulty. Our findings show that LLM performance on this baseline is comparable to the original benchmark and significantly better than on the proposed reasoning-complex tasks. This indicates that the primary factor driving performance decline on MedResEval is the increased reasoning difficulty, rather than a simple generalization effect caused by question reformulation. Thank you again for your valuable feedback. More detailed information about this baseline is provided in Appendix A, and we will also further refine the presentation in the main text to more clearly convey our intuition.

---

### Official Review · Reviewer_QwZj · 2024-10-31

**Soundness:** 2
**Presentation:** 3
**Contribution:** 2
**Rating:** 3
**Confidence:** 3

**Summary:**

The paper investigates the impact of training corpus size and model parameter size on the performance of large language models (LLMs) in the medical domain. The authors introduce a new benchmark, MedResEval, which is designed to demand more complex reasoning and decision-making, reflecting real-world medical scenarios more accurately. Through comprehensive analysis, the paper reveals that while smaller models can approach the performance of larger models on simple tasks, they underperform on complex tasks requiring advanced reasoning. The authors also develop a difficulty-dependent scaling-law formula to characterize the performance of LLMs with varying training data sizes at a fixed model parameter size. The study emphasizes the importance of model parameter scales, particularly for complex reasoning tasks, and suggests that sufficiently large parameters are essential for effectively addressing real-world medical scenarios.

**Strengths:**

1. The paper provides a novel analysis of the scaling effects of LLMs within the medical domain, an area critical for the application of advanced reasoning capabilities. The creation of MedResEval, a benchmark requiring complex reasoning, is a contribution as it allows for more accurate assessment of LLMs in medical scenarios.
2. The paper is well-structured, with a clear problem formulation and methodology. The experiments are thorough, involving multiple LLM families and a range of model sizes and training data amounts. The analysis include both qualitative and quantitative assessments.
3. The paper is also well-written and easy to follow. The introduction of the problem, related work, methodology, experiments, and results are clearly presented. The use of figures and tables to summarize the study's process and findings is effective.
4. The study's findings are significant as they provide insights into the limitations of current LLMs in handling complex reasoning tasks, which is crucial for their deployment in high-stakes domains like healthcare. The proposed scaling-law formula offers a predictive tool for future model development.

**Weaknesses:**

1. Generalizability: While the paper focuses on the medical domain, it's unclear how these findings generalize to other domains requiring complex reasoning. Further discussion on the broader implications of these results would be beneficial. When extended to other domains, the conclusions may change.
2. Data Diversity: The paper primarily uses one benchmark (MedQA) as the basis for MedResEval. It would be valuable to see how the models perform on other medical datasets to ensure the results are not dataset-specific. At the same time, the so-called "more complex" tasks are not expanded enough, and more complex medical scenario problems should be designed.
3. Model Diversity: The study focuses on a limited number of LLM families. Including a more diverse set of models, including those with different architectures, could provide a more comprehensive understanding of the scaling effects.

**Questions:**

1. Generalization to Other Domains: How do the authors envision their findings generalizing to other domains that require complex reasoning, such as legal or financial analysis?
2. Impact of Data Diversity: Are there any plans to validate the findings using other medical datasets to ensure the results are not specific to MedQA?
3. Model Architecture Variation: Would the inclusion of models with different architectures change the observed scaling effects, and is this something the authors have considered in their analysis?
4. Practical Implications: What are the practical implications of these findings for the deployment of LLMs in clinical settings? How might these insights inform the development of future LLMs for healthcare applications?

---

> ### Author Response · Authors · 2024-12-04
> **Response to Reviewer QwZj (1/2)**
>
> We are sincerely grateful for your detailed and constructive feedback. Below, we provide our responses to each of the concerns you raised.
>
> 1. **Generalizability**: We sincerely appreciate your insightful suggestions. In fact, prior to conducting this study, we observed that some of the latest small-scale LLMs (e.g., LLaMA3-8B) achieved comparable or even superior performance to larger-scale LLMs across various benchmarks, including famous medical benchmarks like MedQA. The remarkable performance of these small-scale LLMs has led some researchers to believe that small-scale LLMs trained on larger-scale datasets can rival the performance of larger-scale LLMs. The primary goal of our study is to empirically analyze the effects of training data size $N$ and model parameter size $D$ on LLMs’ reasoning abilities by comparing the performance of a series of LLMs on tasks with varying reasoning difficulty. Our evaluation results reveal that while the latest small-scale LLMs achieve performance comparable to larger-scale LLMs on simpler tasks, their performance falls significantly behind on more challenging reasoning tasks. This indicates that for complex reasoning tasks, merely scaling up the training data size is insufficient; scaling up the model parameter size is also essential. The conclusions drawn from this study provide valuable guidance for the future development and application of LLMs in reasoning tasks.
>
>    Considering that medicine is a key application domain for large language models, and solving problems in this field requires expert-level reasoning and decision-making, we have primarily conducted a case study on the medical domain. While we acknowledge that findings in our paper need further validation to generalize to other domains, the underlying principles of scaling effects and reasoning complexity explored in our work are likely applicable to other reasoning-intensive fields. In the revised paper, we will add a discussion to highlight these points and clarify the potential scope of generalizability.
>
> 2. **Data Diversity**: We are very grateful for your constructive suggestion. Our work focuses on studying the scaling effects of LLMs across tasks with varying reasoning complexities, and MedResEval—derived from the widely recognized MedQA benchmark—was specifically designed for this purpose. The reformulations we introduced significantly increased reasoning difficulty from three key aspects, making it sufficient to investigate the scaling effects of LLMs in medical reasoning. While the proposed MedResEval is sufficient for demonstrating our claims, we plan to further incorporate other medical datasets in the future to further validate the generalizability of our claims.
>
> 3. **More Complex Tasks are Not Expanded Enough**: Thank you for your thoughtful suggestions. The primary goal of this work is to empirically analyze the effects of training data size ($N$) and model parameter size ($D$) on LLMs’ reasoning abilities by comparing the performance of a series of LLMs on tasks with varying reasoning difficulties. To control for other variables as much as possible (e.g., directly adopting other reasoning-complex medical datasets might introduce variations in medical knowledge being tested, potentially affecting the fairness of the study), we derived our evaluation dataset by reformulating existing medical benchmarks based on three key factors to create new tasks with increased reasoning difficulty. Indeed, investigating LLMs’ performance on more complex medical scenario problems is also crucial. Considering this, we also designed a diagnostic simulation task in this work, where we perturbed key diagnostic information and required LLMs to reconstruct the correct diagnostic details and complete the corresponding diagnostic task. While our current MedResEval dataset is sufficient for studying the scaling effects of LLMs across medical tasks with varying reasoning complexities, we also plan to extend our evaluation to include more complex medical scenario problems to further evaluate the medical reasoning capabilities of current LLMs.

---

> > ### Author Response · Authors · 2024-12-04
> > **Response to Reviewer QwZj (2/2)**
> >
> > 4. **Model Diversity**: Thank you for your thoughtful suggestions. In our work, we primarily conduct our analysis based on the evaluation across a total of 12 base models from two LLM families (Qwen and Llama). These two families represent the most popular open-source LLM families at present, and a significant body of research has already been conducted around them. Indeed, in our early investigations, we also considered including more LLMs from other families to provide a more comprehensive analysis. However, other LLM families (e.g., Gemma) generally only cover a very limited range of parameter scales (typically < 10B) and lack open-source models at scales of 70B or larger. This limitation makes these families less suitable for supporting our study on scaling effects.
> > 5. **Practical Implications**: Thank you for your insightful comments. Our research highlights that while both data and parameter scales enhance LLM performance, greater emphasis must be placed on parameter scales, particularly for complex reasoning tasks. Therefore, when training LLMs for real-world clinical scenarios, it is essential to perform medical-specific fine-tuning on sufficiently large general-domain LLMs (e.g., $\geq$70B) to ensure their reasoning capabilities are adequate for handling complex tasks in real clinical contexts. Directly fine-tuning small-scale LLMs (e.g., Llama3-8B) for the medical domain is unlikely to achieve sufficient medical reasoning ability.

---

### Official Review · Reviewer_pia3 · 2024-11-03

**Soundness:** 2
**Presentation:** 3
**Contribution:** 2
**Rating:** 5
**Confidence:** 2

**Summary:**

The paper proposes a new benchmark dataset, MedResEval, built over the MedQA dataset, by varying 3 dimensions of difficulty. The authors also propose a difficulty dependent scaling law and results for the same with general purpose LLMs. They tackle the question of whether smaller LLMs can do as well as larger LLMs if given sufficiently large datasets, even when difficulty level of the data changes. The authors seek to identify boundaries to the application of smaller LLMs under specific constraints like data difficulty.

**Strengths:**

•	Significance: The authors have taken on a relevant problem, especially given the growing landscape of LLMs in the medical context. The authors propose a relevant benchmark that can aid further research in this area.

•	Quality: The authors have performed a quantitative and qualitative assessment of their dataset. The authors have conducted evaluations with 12-18 open source models from 2 model families.

•	Clarity: The writing is quite clear. The authors have provided good examples to illustrate the modifications added.

•	Originality: The novelty is in the proposed dataset and modification to the scaling law in the event of changing difficulty, although the findings themselves are not completely surprising.

**Weaknesses:**

•	The authors have not shared the proposed dataset yet, which is a key contribution.

•	The main issue is that the evaluation is limited to general purpose LLMs. Since the context is the medical domain, it would be more impactful to examine the effect on the scaling law and the effect of varying difficulty levels on medical LLMs like MedPALM[1], Meditron[2] etc.

•	The authors have only evaluated on MedResEval, which is derived from MedQA. Other medical datasets like MedMCQA[3] or PubMedQA[4] can also be considered. It would also be good to give an intuition of how these can be modified to increase the difficulty levels.

[1] Singhal K, Tu T, Gottweis J, Sayres R, Wulczyn E, Hou L, Clark K, Pfohl S, Cole-Lewis H, Neal D, Schaekermann M. Towards expert-level medical question answering with large language models. arXiv preprint arXiv:2305.09617. 2023 May 16.

[2] Chen Z, Cano AH, Romanou A, Bonnet A, Matoba K, Salvi F, Pagliardini M, Fan S, Köpf A, Mohtashami A, Sallinen A. Meditron-70b: Scaling medical pretraining for large language models. arXiv preprint arXiv:2311.16079. 2023 Nov 27.

[3] Pal A, Umapathi LK, Sankarasubbu M. Medmcqa: A large-scale multi-subject multi-choice dataset for medical domain question answering. InConference on health, inference, and learning 2022 Apr 6 (pp. 248-260). PMLR.

[4] Jin Q, Dhingra B, Liu Z, Cohen WW, Lu X. Pubmedqa: A dataset for biomedical research question answering. arXiv preprint arXiv:1909.06146. 2019 Sep 13.

**Questions:**

•	The intuition behind Eq 3 and how it relates to Eq 1 can be elaborated on, to aid readers.

•	In Eq 3, is the difficulty dependent aspect only coming from the separation into $i$ tasks? If not, please elaborate.

•	In line 179, it would be good to highlight why the 3 aspects mentioned were the way to increasing difficulty. If possible, please add citations supporting each dimension.

•	The authors have validated the diagnosis simulation task with clinicians. Can a similar evaluation be done for the other 3 tasks and dimensions, to ensure that the questions generated are non-trivial (for example adding relevant options while expanding the decision space)?

•	A complete evaluation should include medical LLMs as well.

•	The authors have not included limitations of the work.

•	In Appendix B, where the 5-shot setting is described, please add in details of the difficulty level of the examples used in 5-shot learning.

•	Minor Comment: A few typos are present in the current draft (eg: sematic instead of semantic in Figure 5)

---

> ### Author Response · Authors · 2024-12-03
> **Response to Reviewer pia3 (1/2)**
>
> We are sincerely grateful for your detailed and constructive feedback. Below, we provide our responses to each of the concerns you raised.
>
> 1. **Data Availablity**: Thank you for your constructive comments. We will make all the datasets and codes involved in this paper publicly available to facilitate the related research.
>
> 2. **Analysis on Medical LLMs**: Thank you very much for your insightful suggestions. The primary goal of our work is to revisit the scaling effects of LLMs by analyzing how training corpus size and parameter size influence model performance on problems of varying reasoning difficulty. Given that medicine is a critical application domain for LLMs and solving medical problems heavily relies on reasoning abilities, we conducted a case study in the medical domain. Considering the current limited number and variety of medical LLMs, as well as the differences in their domain-specific fine-tuning procedures, our paper primarily focuses on evaluating and analyzing general-domain LLMs. Based on your constructive feedback, we also conducted a preliminary study by evaluating the Meditron series models (since MedPALM has not been open-sourced, we are currently unable to evaluate it but will do so when possible). Meditron-7B/70B are medical LLMs fine-tuned from Llama2, while Meditron3-8B/70B are fine-tuned from Llama3. The experimental results are as follows:
>
>    | Model         | Simple | Complex |
>    | ------------- | ------ | ------- |
>    | Meditron-7B   | 11.6   | -2.4    |
>    | Meditron-70B  | 40.8   | 12.8    |
>    | Meditron3-8B  | 44.8   | 10.5    |
>    | Meditron3-70B | 65.9   | 33.9    |
>
>    We found that medical LLMs exhibited trends similar to general-domain LLMs across tasks of varying reasoning difficulty. On simpler tasks, Meditron3-8B outperformed Meditron-70B by 4%. However, on more complex tasks, Meditron-70B still surpassed Meditron3-8B by over 2%. This suggests that medical LLMs are also consistent with the conclusions presented in our paper.
>
> 3. **Involving More Medical Datasets**: We sincerely appreciate the suggestion to incorporate other datasets like MedMCQA or PubMedQA. Our work focuses on studying the scaling effects of LLMs across tasks with varying reasoning complexities, and MedResEval—derived from the widely recognized MedQA benchmark—was specifically designed for this purpose. The reformulations we introduced significantly increased reasoning difficulty from three key aspects, making it sufficient to investigate the scaling effects of LLMs in medical reasoning. Once again, we thank you for the valuable suggestion and plan to explore the inclusion of additional datasets (e.g., MedMCQA) in the future to further validate the extensibility of our evaluation framework.
>
> 4. **Intuition Behind Eq3**: Thank you for your valuable suggestion. The intuition behind Eq3 primarily stems from observations of Eq1, which indicates that when the parameter size N and computational cost C are fixed, the training data size D and test loss L exhibit a power-law relationship. Considering that downstream task performance P is negatively correlated with test loss L, we hypothesize that P may also share a similar power-law relationship with D, where P improves with increasing D before eventually saturating. We further empirically validated this hypothesis through experiments. We will elaborate further on this intuition in the revised version of the paper.
>
> 5. **Difficulty Dependent Aspect of Eq3**: We greatly appreciate your constructive suggestion. The difficulty-dependent aspect in Eq3 is not solely captured by the separation into tasks. It is also reflected in the parameter $\alpha_{N,i}$, which varies with both the model size $N$ and the specific downstream task $i$. This parameter encapsulates the inherent complexity of the task and how a model with a specific parameter size $N$ handles it, thus directly influencing the performance scaling behavior. Thank you for your suggestion, and we will elaborate this in our revised paper.
>
> 6. **Highlighting why the 3 aspects mentioned were the way to increasing difficulty**: Thank you for your constructive feedback. We have provided a detailed explanation of how these three aspects influence reasoning difficulty in Section 3.2 (line 181- line 196). Based on your valuable suggestions, we will further refine the relevant descriptions to enhance clarity in our revised paper.

---

> > ### Author Response · Authors · 2024-12-03
> > **Response to Reviewer pia3 (2/2)**
> >
> > 7. **Human Evaluation on Other 3 Tasks**: Thank you for your thoughtful suggestion. When designing the diagnosis simulation task, we perturbed key diagnostic information (e.g., gender, symptoms, test results) to evaluate whether LLMs can resolve contradictions in the diagnostic information and complete the diagnosis based on the revised questions. However, perturbing these key diagnostic information may lead to scenarios where the answer changes, becomes non-existent in options, or remains unaffected. Therefore, we engaged doctors to validate the perturbed questions, essentially re-annotating these reformulated questions. For the expanding-decision-space question type, we also had doctors validate the generated distractors to ensure they are not correct answers to the original questions. For the other two tasks, the construction process is more deterministic, and the labels for the adapted questions can be derived directly from the construction process. Once again, we sincerely appreciate your suggestion, and we will include additional details in the Methods section of the paper to enhance clarity.
> >
> > 8. **Evaluating Medical LLMs**: Thank you very much for your insightful suggestions. The primary goal of our work is to revisit the scaling effects of LLMs by analyzing how training corpus size and parameter size influence model performance on problems of varying reasoning difficulty. Given that medicine is a critical application domain for LLMs and solving medical problems heavily relies on reasoning abilities, we conducted a case study in the medical domain. Considering the current limited number and variety of medical LLMs, as well as the differences in their domain-specific fine-tuning procedures, our paper primarily focuses on evaluating and analyzing general-domain LLMs. Based on your constructive feedback, we also conducted a preliminary study by evaluating the Meditron series models, which are finetuned on medical corpora. Meditron-7B/70B are medical LLMs fine-tuned from Llama2, while Meditron3-8B/70B are fine-tuned from Llama3. The experimental results are as follows:
> >
> >    | Model         | Simple | Complex |
> >    | ------------- | ------ | ------- |
> >    | Meditron-7B   | 11.6   | -2.4    |
> >    | Meditron-70B  | 40.8   | 12.8    |
> >    | Meditron3-8B  | 44.8   | 10.5    |
> >    | Meditron3-70B | 65.9   | 33.9    |
> >
> >    We found that medical LLMs exhibited trends similar to general-domain LLMs across tasks of varying reasoning difficulty. On simpler tasks, Meditron3-8B outperformed Meditron-70B by 4%. However, on more complex tasks, Meditron-70B still surpassed Meditron3-8B by over 2%. This suggests that medical LLMs are also consistent with the conclusions presented in our paper.
> >
> > 9. **Limitations of the Work**: We sincerely appreciate your constructive feedback. While this study provides a detailed evaluation of existing LLMs and reveals the impact of training corpus sizes and parameter sizes on the medical reasoning capabilities of these models, it has the following limitations: (1) Although this study includes 10+ LLMs from two popular LLM families (Llama and Qwen), due to the current availability of open-source LLMs, our evaluation focuses primarily on models with parameter sizes around ~7B and ~70B and pays less attention on intermediate-sized models, as such open-source models are less readily available; (2) While this study conducts a case study in the medical domain and investigates the difficulty-dependent scaling law of LLM reasoning capabilities, this methodology has the potential to be extended to other reasoning-intensive domains. In future study, we plan to include more models of other sizes when research conditions permit, and conduct experiments in other reasoning-intensive domains to further validate the generalizability of our conclusions. We will incorporate these limitations and future work into the revised version of our paper.
> >
> > 10. **Details of the Difficulty Level of the Examples Used in 5-shot Learning**: Thank you for your constructive feedback. In our evaluation, to maximize the performance of LLMs, we selected five examples for each question type to perform 5-shot learning, ensuring that the difficulty level of the demonstrative examples matched that of the test samples. Based on your valuable suggestion, we will further elaborate on the details of our 5-shot learning evaluation process in Appendix B, including the specific 5-shot learning examples used for each question type.
> >
> > 11. **Few Typos**: We sincerely appreciate your detailed review. In our revised paper, we will carefully proofread to eliminate these typos.

---

> > > ### Comment · Reviewer_pia3 · 2024-12-03
> > >
> > > Thank you for the detailed response and for running the experiment on Meditron. However, I will retain my current score. At this stage my chief concern is still about whether the findings will generalise well to all medical LLMs. I do understand that it may be difficult to run experiments on other medical LLMs (for example OpenBioLLM) at this point in the discussion phase, but I believe this may be needed to substantiate your conclusions as the focus is on a specific domain.

---

### Official Review · Reviewer_Gb62 · 2024-11-04

**Soundness:** 3
**Presentation:** 4
**Contribution:** 3
**Rating:** 6
**Confidence:** 3

**Summary:**

The paper proposed a new benchmark for LLMs' medical reasoning capabilities, MedResEval, built on the MedQA dataset. MedResEval is designed to evaluate the scaling effects of LLMs on medical reasoning capabilities from training corpus sizes and parameter sizes (or model size). From various evaluation experiments and in-depth results analysis, the paper concluded that both training data size and parameter scales would enhance LLM performances on medical reasoning, and parameter scales lead to a more pronounced performance improvement than scaling training data size for complex reasoning tasks.

**Strengths:**

- Originality: the paper proposed a novel benchmark to evaluate the utility of LLMs on medical reasoning by expanding the existing popular MedQA dataset with more complex question representations, larger decision space and multi-step tasks.
- Quality: the paper is solid in technical soundness with meaningful experiment design and proposed evaluation metrics that fit the hypotheses to test the scaling factors of different LLMs. Most conclusions are based on quantitive performance comparison.
- Clarity: the paper is well-structured with good illustration of diagrams, plots and tables.
- Significance: the benchmark proposed by the paper is a meaningful expansion of the existing popular MedQA dataset. Also, the same approach could also be applied to other medical benchmark datasets like MedMCQA, PubMedQA, etc. Also, the scaling factor of LLMs is an interesting and important question on practical utility of LLMs in medical domain. The paper offers a good insight or framework on carefully examination of the marginal gain/loss of increasing training data or parameters.

**Weaknesses:**

- Lack of limitations and future work in Conclusion part.
- The bar plots somehow are a little bit hard to illustrate the performance changes by various Ns & Ds. Scatter plots like Figure 6 (with dot sizes indicating N or D) might work better.
- It might be better to indicate both x-axis and y-axis are in log-scale in Figure 6 caption.
- Overall, the performance differences lack significant analysis since only average performance is reported (e.g. Figure 5). Pls add confidence intervals if they are available.

**Questions:**

- Will the new benchmark be published, including the diagnosis case study?
- In page 5 "Expanding Decision Space": how to make sure that the randomly generated answers are wrong? Assuming correct answers might be also generated.
- Does the proposed benchmark support multi-answer questions? (2 or more answers are correct)
- It seems the "reasoning steps" only add an additional intermediate task but still towards the same end task. Taking more steps to achieve the same correct answer seems to be a disadvantage instead of advantage. Should this be rephrased as additional task?

---

> ### Author Response · Authors · 2024-12-02
> **Response to Reviewer Gb62 (1/2)**
>
> We are sincerely grateful for your detailed and constructive feedback and very sorry for the late response. Below, we provide our responses to each of the concerns you raised.
>
> 1. **Lack of limitations and future work in Conclusion part**: We sincerely appreciate your constructive feedback. While this study provides a detailed evaluation of existing LLMs and reveals the impact of training corpus sizes and parameter sizes on the medical reasoning capabilities of these models, it has the following limitations: (1) Although this study includes 10+ LLMs from two popular LLM families (Llama and Qwen), due to the current availability of open-source LLMs, our evaluation focuses primarily on models with parameter sizes around ~7B and ~70B and pays less attention on intermediate-sized models, as such open-source models are less readily available; (2) While this study conducts a case study in the medical domain and investigates the difficulty-dependent scaling law of LLM reasoning capabilities, this methodology has the potential to be extended to other reasoning-intensive domains. In future study, we plan to include more models of other sizes when research conditions permit, and conduct experiments in other reasoning-intensive domains to further validate the generalizability of our conclusions. We will incorporate these limitations and future work into the revised version of our paper.
>
> 2. **Bar plots are a little bit hard to illustrate the performance changes by various Ns & Ds**: Thank you very much for your constructive suggestions. In the paper, we primarily used bar plots to present the experimental results, aiming to intuitively compare the performance gaps between the latest small-scale LLMs and larger-scale LLMs across tasks with varying reasoning difficulties. Based on your suggestions, we will revise the presentation of this section in the updated manuscript to provide a clearer comparison of performance changes across different Ns and Ds.
>
> 3. **Indicating both x-axis and y-axis are in log-scale in Figure 6 caption**：We are very grateful for your detailed suggestions. We will revise the caption of Figure 6 to help readers better understand the experimental results.
>
> 4. **Significant Analysis Issue**: We are very grateful for your constructive feedback. To ensure the robustness of our conclusions, we employed the self-consistency method during model evaluation. Specifically, we prompted the LLMs to generate answers to the same question five times and then determined the final model prediction using a majority vote. This approach effectively enhances the performance of LLMs and contributes to the robustness of the evaluation results. Based on your kind suggestions, we also conducted the significant analysis by conducting two additional repeated experiments. Given the high computational cost of evaluating all models, we prioritized supplementary experiments on the four models from the Llama family. The results of these experiments, with 95% confidence intervals, are as follows:
>
>    | Task                       | llama2-7B | llama2-70B | llama3-8B | llama3-70B |
>    | -------------------------- | --------- | ---------- | --------- | ---------- |
>    | Original                   | 16.2±0.3  | 41.8±1.3   | 44.8±0.5  | 64.9±0.0   |
>    | Reducing Available Clues   | 0.8±0.1   | 17.9±0.2   | 13.4±0.7  | 29.2±0.8   |
>    | Expanding Decision Space   | 0.5±0.3   | 6.9±0.6    | 0.9±0.2   | 28.7±0.6   |
>    | Increasing Reasoning Steps | -1.3±0.1  | 26.0±1.2   | -1.2±0.2  | 37.5±0.4   |
>
>    We observed that the confidence intervals obtained from multiple independent experiments are relatively small since our testing setting inherently enhances robustness. Based on your valuable feedback, we will incorporate this significance analysis into the results presented in the paper to further demonstrate the robustness of our findings.

---

> > ### Author Response · Authors · 2024-12-02
> > **Response to Reviewer Gb62 (2/2)**
> >
> > 5. **Data Availability Issue**: We are very thankful for your comments. We will make all the datasets and codes involved in this paper publicly available to facilitate the related research.
> > 6. **Ensuring the Generated Distractors are Wrong**: Thank you for your valuable feedback. This is an important issue, and we have also considered this in our experiments. On one hand, we automatically calculate the similarity score between the generated distractor and the correct option, and if the distractors were highly similar to the correct option, a new distractor was regenerated. On the other hand, we engaged a doctor with five years of experience to further verify whether the generated distractors could potentially be correct answers to the original question.
> > 7. **Multi-answer question Support**:Thank you for your constructive suggestions. The primary goal of this work is to analyze the scaling effects of LLMs on tasks with varying reasoning complexities. To this end, we focused on three factors influencing reasoning difficulty and designed three types of evaluation tasks for our study. Theoretically, the proposed benchmark can support various tasks to evaluate different aspects of medical reasoning capabilities in LLMs. For instance, incorporating multi-answer questions, as you suggested, could prevent LLMs from leveraging the prior knowledge that “only one option is correct.” Instead, LLMs would need to evaluate the feasibility of each candidate option individually to arrive at the final answer. Of course, the proposed benchmark may require additional refinement to effectively support multi-answer questions (e.g., generating extra correct options). We plan to introduce more evaluation tasks in future work to further enhance our framework.
> > 8. **"Reasoning steps" only add an additional intermediate task but still towards the same end task**: Thank you very much for your thoughtful feedback. When designing the *increasing-reasoning-step* task, we added an additional intermediate task (judging whether a given answer is correct) to the original MCQ task, thereby increasing the number of reasoning steps required. Our goal here is to evaluate the ability of LLMs to solve tasks with more reasoning steps. Indeed, from a problem-solving perspective, reaching the same correct answer through more steps can be seen as a disadvantage. However, from an evaluation perspective, this design allows us to isolate and analyze the impact of increasing reasoning steps towards LLMs' performance. For a human with normal-level reasoning capabilities, if the original MCQ task is answered correctly, they can easily determine the intermediate task result by comparing the given answer with their prediction of the original MCQ. However, in our experiments, we observed that the current LLMs performed significantly worse on the *increasing-reasoning-step* task compared to the original MCQ task, indicating that current LLMs are generally less effective than humans at handling multi-step reasoning. Moreover, the performance of 7B models was substantially lower than that of 70B models, which also aligns with the core conclusions of our paper.

---

### Official Review · Reviewer_p5VX · 2024-11-04

**Soundness:** 2
**Presentation:** 3
**Contribution:** 2
**Rating:** 6
**Confidence:** 3

**Summary:**

This paper introduces MedResEval, an evaluation framework designed to examine the impact of model parameters and dataset size on the performance of large language models (LLMs) across four specified tasks. The framework defines a formula based on neural scaling law that models the relationship between performance, parameter count, and dataset size, closely aligning with empirical findings. However, there are concerns about the clinical rigor of the MedResEval framework, as it generates a "complex" dataset with certain definitions that may not fully align with established clinical insights.

**Strengths:**

The paper presents an in-depth evaluation of the proposed MedResEval framework, specifically testing the effects of $N$ (number of parameters) and $D$ (dataset size) — the critical elements of the scaling law. The study defines a formula that effectively models the relationship between performance, parameter count, and dataset size, aligning well with empirical results.

**Weaknesses:**

Although MedResEval introduces a new evaluation framework with results that adhere to a defined scaling rule, concerns persist about its clinical relevance, and some claims regarding its clinical rigor appear overstated.

1. The task definitions in Section 3.2 somewhat overstate the clinical relevance and how each task contributes to the complexity of clinical questions.
- Available Clues: If the answer provided within the paragraphs (as in Figure 8) includes an obviously correct or easily dismissible wrong answer, this could reduce the complexity of the original MCQ. In many challenging MCQs, the difficulty lies in choosing between two or three closely related options. The example in Figure 8 suggests that the LLM only needs to determine if the single integrated answer choice is correct, which may simplify the question.
- Decision Space: Including an easily dismissible wrong answer does not necessarily increase the complexity of the question. Maintaining question complexity would require distractors that present a closer challenge, as straightforward wrong options may not sufficiently elevate the complexity of decision space.
- Reasoning Steps: Verifying whether a randomly provided answer is correct could simplify the task, as the model only needs to evaluate a single option rather than considering multiple potential answers, thus reducing the overall complexity.

2. The evaluations lack confidence intervals, which weakens the robustness and reliability of the claims presented in this paper.

Although the presentation and evaluation of the paper were quite comprehensive, this limitation is viewed to be critical and hard to fix at this point of submission. Because this limitation would reduce the impact and contribution of the paper to medical applications, I am inclined to reject the paper in its current form. However, if there could be any improvements that could be made in the short term that address this concern, would be open to revisiting this decision.

**Questions:**

Does $K$ in Equation 2 refer to the number of tasks considered in MedResEval (specifically, the 4 tasks)? It would be helpful if this were clearly indicated in the manuscript.

---

> ### Author Response · Authors · 2024-12-02
> **Response to Reviewer p5VX (1/3)**
>
> We are sincerely grateful for your detailed and constructive feedback and very sorry for the late response. Below, we provide our responses to each of the concerns you raised.
>
> 1. **Clinical Relevance of Questions in MedResEval**: We sincerely appreciate your insightful comments. In fact, prior to conducting this study, we observed that some of the latest small-scale LLMs (e.g., LLaMA3-8B) achieved comparable or even superior performance to larger-scale LLMs across various benchmarks, including famous medical benchmarks like MedQA. The remarkable performance of these small-scale LLMs has led some researchers to believe that small-scale LLMs trained on larger-scale datasets can rival the performance of larger-scale LLMs. The primary goal of our study is to empirically analyze the effects of training data size $N$ and model parameter size $D$ on LLMs’ reasoning abilities by comparing the performance of a series of LLMs on tasks with varying reasoning difficulty. Our evaluation results reveal that while the latest small-scale LLMs achieve performance comparable to larger-scale LLMs on simpler tasks, their performance falls significantly behind on more challenging reasoning tasks. This indicates that for complex reasoning tasks, merely scaling up the training data size is insufficient; scaling up the model parameter size is also essential. The conclusions drawn from this study provide valuable guidance for the future development and application of LLMs in reasoning tasks.
>
>    Considering that medicine is a key application domain for large language models, and solving problems in this field requires expert-level reasoning and decision-making, we have primarily conducted a case study on the medical domain. We found that real-world medical reasoning tasks are often far more complex than questions in LLM benchmarks. In this work, we specifically focus on three key factors that influence reasoning difficulty:
>
>    1. **Available Clues**: Questions in medical benchmarks often provide additional reasoning clues (e.g., for multiple-choice questions (MCQs), LLMs can leverage elimination strategies to rule out incorrect multiple-choice options). In contrast, real-world medical tasks typically involve less available clues. For example, in clinical diagnosis tasks, doctors must derive the diagnosis based solely on the given patient information, without relying on options to guide reasoning.
>
>    2. **Decision Space**: Questions in medical benchmarks usually have a small decision space (e.g., 4–6 options for MCQ), whereas real-world medical tasks often require reasoning within a much larger decision space. For instance, in clinical diagnosis, potential decision options may include hundreds or more candidate diseases.
>
>    3. **Reasoning Steps**: Medical benchmark questions typically require single-step reasoning, while real-world medical tasks often involve multi-step reasoning. For example, a doctor must first arrive at a clinical diagnosis and then determine the appropriate treatment plan based on the diagnosis.
>
>    Indeed, the three key factors discussed in this study are not only closely tied to medical reasoning but are also highly relevant to other reasoning-intensive domains. Therefore, our findings have the potential to be extended to these areas, providing insights for LLM reasoning research. Thank you for your feedback; we will further clarify the motivation and clinical relevance of our work in the paper.

---

> ### Author Response · Authors · 2024-12-02
> **Response to Reviewer p5VX (2/3)**
>
> 2. **Complexity of Questions in MedResEval**: Thank you for your thoughtful comments. In this study, we designed three tasks targeting the three key factors you mentioned, generating three types of questions based on existing medical benchmarks. The issues you pointed out are indeed critical, and we have carefully considered them during the dataset creation process, implementing corresponding designs to address them:
>
>    1. **Reducing Available Clues**: As you pointed out, the difficulty of multiple-choice questions (MCQs) lies in selecting the correct answer from several closely related options. Considering this, we constructed the evaluation samples by taking each original MCQ and pairing the correct option with a selected wrong option to generate two statement verification questions with opposite labels. LLMs must correctly evaluate the validity of both statements independently to perform better than random guessing, without relying on hints by comparing between options. This ensures that LLMs evaluate the correctness of both the true and misleading options without contextual clues from comparing choices.
>
>    2. **Expanding Decision Space**: Indeed, including an easily dismissible incorrect answer does not necessarily increase the complexity of the question. To tackle this, we **replaced the original correct answer with a distractor**, creating another MCQ that does not have a correct answer in the options. Then, we provide both the original and modified questions to LLMs, requiring them to **determine whether a correct answer exists among the given options**. This design significantly expands the decision space, as LLMs must not only evaluate the given options but also consider the possibility that none of them is correct.
>
>    3. **Increasing Reasoning Steps**: Of course, evaluating the correctness of a single provided option may reduce reasoning complexity. We have also considered this important point and retained the original MCQ question along with all its options. Actually, we introduced a two-step reasoning task: LLMs are first asked to determine whether a given option is correct for the MCQ. If the answer is incorrect, they must then identify the correct option from the rest options. For example:
>
>       > **Question:** A 25-year-old man comes to the office because of pain in his left shoulder… Which of the following enzymes is most likely deficient in this patient?
>       >
>       > **Options**:
>       >
>       > **A:** Branched-chain alpha-ketoacid dehydrogenase
>       >
>       > **B:** Cystathionine synthase deficiency
>       >
>       > **C**: Homogentisic acid oxidase
>       >
>       > **D:** Phenylalanine hydroxylase
>       >
>       > **E:** Propionyl-CoA carboxylase
>       >
>       > **Alice chose answer D.** Please verify if this is correct, and if not, provide the correct answer.
>       >
>       > **Answer:** incorrect, answer C is the correct answer
>
>    To answer such questions, LLMs need to not only evaluate whether Alice’s choice is correct but also identify the correct answer if it is not. Similar to the reducing-available-clues task, we also generated two questions for each original MCQ by pairing the correct option with a distractor. This ensures a more comprehensive evaluation of the LLMs’ reasoning abilities.
>
> Thank you once again for your valuable feedback. We will further refine our paper to enhance the clarity of our implementation details.

---

> > ### Author Response · Authors · 2024-12-02
> > **Response to Reviewer p5VX (3/3)**
> >
> > 3. **Confidence Intervals Issue**: We are very grateful for your constructive feedback. To ensure the robustness and reliability of our conclusions, we employed the self-consistency method [1] during model evaluation. Specifically, we prompted the LLMs to generate answers to the same question five times and then determined the final model prediction using a majority vote. This approach effectively enhances the performance of LLMs and contributes to the robustness of the evaluation results. Based on your kind suggestions, we also conducted two additional repeated experiments to calculate confidence intervals. Given the high computational cost of evaluating all models, we prioritized supplementary experiments on the four models from the Llama family. The results of these experiments, with 95% confidence intervals, are as follows:
> >
> >    | Task                       | llama2-7B | llama2-70B | llama3-8B | llama3-70B |
> >    | -------------------------- | --------- | ---------- | --------- | ---------- |
> >    | Original                   | 16.2±0.3  | 41.8±1.3   | 44.8±0.5  | 64.9±0.0   |
> >    | Reducing Available Clues   | 0.8±0.1   | 17.9±0.2   | 13.4±0.7  | 29.2±0.8   |
> >    | Expanding Decision Space   | 0.5±0.3   | 6.9±0.6    | 0.9±0.2   | 28.7±0.6   |
> >    | Increasing Reasoning Steps | -1.3±0.1  | 26.0±1.2   | -1.2±0.2  | 37.5±0.4   |
> >
> >    We found that the confidence intervals obtained from multiple experiments are relatively small, primarily because our test setting inherently enhances robustness. Based on your valuable feedback, we will incorporate confidence intervals into the results presented in the paper to further demonstrate the credibility of our findings.
> >
> >    [1] Wang X, Wei J, Schuurmans D, et al. Self-Consistency Improves Chain of Thought Reasoning in Language Models. ICLR 2023.
> >
> > 4. **Meaning of $K$ in Equation 2**: Thank you for your detailed suggestions. $K$ in Equation 2 does refer to the number of tasks in MedResEval. We will make this clearer in our revised paper.

---

### Meta-Review · Area_Chair_zik9 · 2024-12-21

**Metareview:**

This paper has been evaluated by 5 knowledgeable reviewers. Their opinions varied: 3 marginal acceptances, 1 marginal rejection and one straight rejection. It examines the impact of model complexity and data size on the performance of LLMs on Q&A tasks in healthcare domain. The authors provided a rebuttal and engaged with the reviewers. The remaining issues include lack of clarity of how the approach would translate across a broader list of LLMs and broader collection of Q&A benchmark data. Therefore, the potential impact of this work is not readily apparent. It brings however some insights on how to structure benchmarks for evaluating LLMs is domain-specific application scenarios.

**Additional Comments On Reviewer Discussion:**

Two reviewers engaged in a conversation with the authors, and some of them reconsidered their assessments upon reading reviews by others. These discussions did not substantially affect the final assessment of the fitness of this paper in its current form for ICLR.

---

### Decision · Program_Chairs · 2025-01-22

Reject